# Omalizumab and Dupilumab for the Treatment of Bullous Pemphigoid: A Systematic Review

**DOI:** 10.3390/jcm13164844

**Published:** 2024-08-16

**Authors:** Elena Granados-Betancort, Manuel Sánchez-Díaz, Daniel Muñoz-Barba, Salvador Arias-Santiago

**Affiliations:** 1School of Medicine, University of Granada, 18071 Granada, Spain; 2Skin Autoimmune Diseases Unit, Hospital Universitario Virgen de las Nieves, 18002 Granada, Spain; 3Dermatology Department, Hospital Universitairo Virgen de las Nieves, Instituto de Investigación Biosanitaria IBS Granada, 18002 Granada, Spain; 4School of Medicine, Dermatology Deparment, University of Granada, 18071 Granada, Spain

**Keywords:** bullous pemphigoid, omalizumab, dupilumab

## Abstract

**Background**: Bullous pemphigoid (BP) is an autoimmune disease characterized by the appearance of very pruritic subepidermal blisters. It appears mostly in the elderly and is associated with multiple comorbidities, which makes its management and treatment difficult. The purpose of this systematic review is to compile current information on published cases of BP treated with omalizumab (omalizumab) and dupilumab (dupilumab) in order to obtain information on clinical efficacy and safety data available. **Methods**: A literature search of all cases of BP treated with omalizumab/dupilumab published in the literature up to January 2024 was performed using the Pubmed database. After an exhaustive search, a total of 61 studies encompassing 886 patients met the inclusion criteria and were included in the review. **Results**: The majority of patients with BP treated with omalizumab/dupilumab presented a significant improvement in symptomatology, being very safe drugs with minimal side effects. The main limitation of the presented review is the quality of the included studies, most of them being case series or individual cases. The development of studies with a higher level of scientific evidence in the near future would be of great interest. **Conclusions**: Both omalizumab and dupilumab appear to be effective options for treating BP in patients refractory to other pharmacological therapies. They are drugs with a good safety profile and the adverse reactions associated with their use are infrequent and generally mild.

## 1. Introduction

Bullous pemphigoid (BP) is the most common autoimmune blistering disease in adults in developed countries [1]. It is characterized by the formation of autoantibodies against structural proteins of the dermal-epidermal junction and by the appearance of very itchy hives and subepidermal blisters [2].

Its incidence is about 0.2–3 new cases per 100,000 habitants [3], it appears more frequently in elderly patients (over 70 years of age) [4] and is associated with various comorbidities, such as neurological diseases, or other inflammatory diseases such as rheumatoid arthritis [5]. Probably in relation to the greater burden of comorbidities, as well as the clinical manifestations of the disease, the morbidity and mortality of patients with BP is 5 to 6 times higher compared to the general population adjusted for age and sex [1].

The pathogenesis of this disease is defined by an immunological component (IgG and IgE antibodies against hemidesmosomal proteins BP180 and BP230) and an inflammatory component (action of neutrophils and eosinophils that damage the dermo-epidermal junction). The deposit of antibodies in the basement membrane triggers an inflammatory response responsible for the clinical manifestations of the disease [1]. The ultimate cause is unknown, although exposure to certain drugs has been described as an etiological agent in some cases of BP [6].

The diagnosis of BP is based on the combination of clinical, histological, serological and immunofluorescence data. The suspected diagnosis must be clinical and requires a biopsy for histological and immunofluorescence study, as well as a serological evaluation [7,8].

Regarding current treatment of BP, it should be taken into account, that despite the availability of both topical and systemic treatments, such as corticosteroids and immunsupppresive drugs [9,10], the main limitation in the treatment of BP is the presence of side effects, which especially affect the typical patient group with BP, elderly patients and patients with multiple comorbidities.

Omalizumab is a humanized monoclonal antibody that selectively binds to immunoglobulin E. It is indicated for the treatment of severe allergic asthma, chronic spontaneous urticaria, and chronic rhysnosinusitis with nasal polyps. Various case reports have demonstrated the potential usefulness of omalizumab in BP, which could act by inhibiting the IgE-mediated inflammatory cascade, and is also a drug with an excellent safety profile [11]. On the other hand, Dupilumab is a drug that act son the α subunit of the interleukin 4 receptor (IL-4Rα) inhibiting IL-4 and IL-13 signaling. It is approved for the treatment of asthma, nodular prurigo, atopic dermatitis, and chronic rhinosinusitis with nasal polyposis. Currently, there are published cases in which a clinical improvement has been observed, with cessation of pruritus and reduction in blistering in patients with BP [12]. This improvement, as well as the absence of relevant adverse effects, makes dupilumab postulated as a treatment option for BP.

Given the recent evidence of the potential usefulness of omalizumab and dupilumab in the treatment of BP, as well as their excellent safety profile, it is of great interest to synthesize the available scientific evidence on their use in patients with BP, which is the objective of this systematic review. 

## 2. Materials and Methods

### 2.1. Study Design and Objectives

A systematic review was carried out including all the reports on BP treated with the biological drugs omalizumab and dupilumab, with the objective to analyze the common clinical characteristics, systematize the evolution of the disease, collect effectiveness data, as well as available safety data.

### 2.2. Search Strategy

A bibliographic search of all cases published in the literature up to January 2024 was performed using the Pubmed database. The search command used was: ((pemphigoid) OR (bullous pemphigoid)) AND ((omalizumab) OR (dupilumab)). The PRISMA 2020 guidelines for systematic reviews were followed when carrying out this work.

### 2.3. Inclusion and Exclusion Criteria

The search was limited to: (A) Publications on patients with a clinical diagnosis of BP regardless of severity and presentation treated with omalizumab and dupilumab. (B) Any type of epidemiological study (clinical trials, cohort studies, case-control studies, cross-sectional studies and clinical case presentations). (C) Articles written in English and Spanish. Therefore, the following were excluded: (A) Those publications that did not evaluate patients with a diagnosis of BP treated with omalizumab and/or dupilumab. (B) Clinical guidelines, protocols and conference summaries. (C) Publications written in a language other than English and Spanish.

### 2.4. Selection of Studies

A first search was carried out in which the titles and abstracts were reviewed by two researchers (EGB and MSD) of all the studies obtained when applying the search command. Of all those studies that met the inclusion and exclusion criteria, the full text was reviewed, as well as their bibliographic references in search of additional sources. Articles that raised doubts about their inclusion or exclusion were subject to discussion with a third researcher (SAS) until a consensus was reached. Articles considered relevant were included in the present analysis.

### 2.5. Research Questions

The present systematic review attempted to answer the following questions.

What profile do patients with BP treated with biological drugs have?How effective are omalizumab and dupilumab in the treatment of BP?What is the safety and side effect profile of omalizumab and dupilumab in patients with BP?

### 2.6. Variables

To answer these questions, the variables evaluated were:Clinical and sociodemographic variables related to the characteristics of BP in patients treated with omalizumab/dupilumab, as well as the existence of comorbidities and other autoimmune diseases.Variables related to the therapeutic management carried out (treatment administered, dosage).Variables related to the effectiveness of treatment with omalizumab and dupilumab in BP. The rate of complete response of patients under treatment in the assessed studies was collected.Variables related to the safety of treatments.

### 2.7. Assessment of the Quality of the Scientific Evidence

The level of evidence of the studies included in the systematic review was evaluated according to the “Center for Evidence-Based Medicine” (CEBM). The levels of evidence were evaluated as follows:1a: Evidence obtained from systematic reviews or meta-analysis of randomized controlled clinical trials.1b: Evidence obtained from individual randomized controlled clinical trials.2a: Evidence obtained from systematic reviews or meta-analysis of cohort studies.2b: Evidence obtained from individual cohort studies.3a: Evidence obtained from systematic reviews or meta-analysis of case-control studies.3b: Evidence obtained from individual case-control studies.4: Evidence obtained from case series.5: Evidence obtained from expert opinions.

### 2.8. Statistical Analysis

Descriptive statistical techniques were used to evaluate the characteristics of the patients included in the evaluated publications. Continuous variables were expressed as mean and standard deviation. The qualitative variables were expressed based on their absolute and relative frequencies. Statistical analyzes were carried out with the JMP 9.0.1 program (SAS 105 Institute, Cary, NC, USA).

## 3. Results

After an initial search, 148 articles were found. After reviewing the titles and abstracts of each of them, 69 were discarded for not meeting the inclusion criteria. Therefore, 79 articles were completely reviewed, of which 18 articles were finally discarded since 16 of them were not completely accessible and 2 of them did not assess the impact of omalizumab/dupilumab on BP. Therefore, 61 articles were included in the systematic review that included 886 patients (363 treated with omalizumab and 523 treated with dupilumab) (Figure 1). All the information included in the studies can be seen in Appendix A and Appendix B.

### 3.1. Sociodemographic and Clinical Characteristics of Patients with BP Treated with Omalizumab and Dupilumab

A total of 363 patients with BP who were treated with omalizumab off-label were included in the review. The average age of the patients was 66.7 years. Of the studies that included the sex of the patients, the majority were women (44 vs. 36 men) although not all studies reviewed specified the age/sex of the patients. The majority of patients had multiple comorbidities associated with BP (diabetes mellitus, high blood pressure, osteoporosis, obesity, heart disease, chronic kidney disease, other associated autoimmune diseases). Furthermore, several cases developed BP as a consequence of treatment for other pathologies (such as dipeptidyl peptidase 4 inhibitors for diabetes treatment [13]; BP due to oncological treatment with anti-HER-2 drugs) [14]. The majority of patients had received treatment with several first-line drugs for BP prior to the initiation of treatment with omalizumab (corticosteroids, methotrexate), which had not achieved improvement in the disease.

Regarding patients with BP treated with dupilumab, a total of 523 off-label patients were included. The average age was 68.2 years, of which 59 were women and 174 men (although not all studies specified the age/sex of the patient). The majority of patients presented comorbidities typically associated with BP (cancer, immunosuppression, diabetes, heart failure, osteoporosis). In addition, several patients developed BP after starting treatment for another pathology they had (BP triggered by nivolumab for treatment of lung metastases due to melanoma [15], BP induced by pembrolizumab to treat cervical cancer [16]). Most of them were treated with other first-line drugs (corticosteroids or immunosupppresants) which did not achieve control of the disease.

### 3.2. Effectiveness of BP Treatment with Omalizumab/Dupilumab

Regarding the effectiveness of BP treatment with omalizumab and dupilumab, the following data were found (Table 1). Regarding treatment with omalizumab, the majority of patients with BP who were treated with 300 milligrams (mg) of omalizumab achieved complete remission of the disease (76.13%), achieving the disappearance of the characteristic skin lesions, as well as the pruritus. The time from the start of treatment to the improvement of the lesions varied among the cases presented, from two weeks to six months, with no recurrences after suspending omalizumab in most cases. Treatment duration was variable between case reports. In some cases, omalizumab was associated with rituximab, achieving remission of the disease more quickly [17]. Likewise, in many of these patients an improvement in the disease was observed, assessable by various scales such as the visual analogue scale (VAS). Along with this, the levels of IgE, eosinophils and antibodies also decreased anti-BP180, BP230.

Regarding patients with BP who were treated with dupilumab (600 mg induction + 300 mg maintenance), the majority (70.39%) achieved remission of the disease, with control of symptoms and resolution of blistering lesions, this being objective both by scales (EVA, bullous pemphigoid disease area index (BPDAI)) as well as laboratory data where many of them achieved a reduction in Th2 lymphocytes andantibodiesanti-BP180, BP230. The time to achieve improvement ranged from two weeks to six months, after which dupilumab was suspended, maintaining complete remission in most cases. The association of dupilumab with CTC achieved better BP control in certain cases [18,19].

### 3.3. Side Effects of Treatment with Omalizumab/Dupilumab

The majority of patients treated with omalizumab did not experience any adverse effects. Some adverse effects that were observed were: dermatitis at the injection site that resolved spontaneously, thrombocytopenia in two patients (one of them did not need to stop AOM and another of them who had multiple comorbidities died), intense pruritus that resolved by adding dupilumab and exacerbation of skin lesions in a patient who required discontinuation of omalizumab. The majority of patients tolerated the treatment adequately, with adverse effects being infrequent and mostly mild.

Most patients treated with dupilumab did not experience any adverse effects. Some presented eosinophilia (in two patients, it was resolved by adding immunosuppresive drugs), thrombosis on two occasions, dermatitis at the injection site that did not require suspending dupilumab, two patients developed pneumonia in relation to their comorbidities that did not require suspending dupilumab and were cured with antibiotics. As in the case of omalizumab, most of these adverse effects were mild and transient. 

## 4. Discussion

Omalizumab and dupilumab are two biological drugs widely used in dermatology for the treatment of pathologies such as chronic spontaneous urticaria and atopic dermatitis [1]. In light of the results of the present systematic review, they could be useful in the management of BP, given their effectiveness and safety data.

Regarding BP, both treatments, omalizumab and dupilumab could act on the pathogenesis of the disease. On the one hand, omalizumab acts as an anti-IgE drug, blocking anti-hemidesmosomal IgE antibodies which are involved in the development of the inflammatory reaction. On the other hand, dupilumab blocks IL-4 and IL-13 action, therefore inhibiting the Th2 pathway which is overexpressed in BP lesions [1]. 

The patients analyzed in the present systematic review who have received treatment with omalizumab and dupilumab are mostly elderly patients, with multiple comorbidities [4]. This study population resembles the patient profile commonly seen in real clinical practice [5], which adds value to the results obtained, facilitating the translation of the results of the review to medical practice. However, it should be taken into account that patients treated with omalizumab and dupilumab are mainly refractory to other treatments, which is why these drugs were administered off-label. The evaluation of the effectiveness and safety of the early treatment with biologic drugs for BP patients could be of interest to avoid the use of immunosuppressive drugs in patients with an increased comorbidity and mortality. 

Regarding effectiveness data, omalizumab and dupilumab could be considered effective drugs for BP, with complete response rates of 76.13% and 70.39% in the reviewed literature. These data are better than response rates seen in other studies for drugs such as methotrexate [20] or oral corticosteroids [21]. On the other hand, the dosage of omalizumab (300 mg subcutaneous every four weeks) and dupilumab (600 mg subcutaneous induction + 300 mg subcutaneous every one to two weeks) could favor greater adherence to treatment in elderly patients. Directly observed treatment (administered by nursing or qualified personnel) may even be useful [18,22]. Although exact complete response rates are not totally comparable due to the lack of standardized criteria among the included studies, future information on the clinical characteristics which could act as biomarkers of response to these drugs are necessary. The predominant implication of IgE antibodies or Th2 inflammatory response in each patient could make a difference in terms of clinical response.

On the other hand, it must be taken into account that the main limitation found in routine clinical practice for the treatment of BP is the comorbidity that patients present, in addition to their advanced age. This fact greatly limits the use of drugs that could be effective, such as systemic corticosteroids or immunosuppressive drugs, but which have an unfavorable side effect profile. Side effects found in the literature reviewed for omalizumab include injection site dermatitis, persistent pruritus, exacerbation of skin lesions, and thrombopenia. In the case of dupilumab, the most frequently described side effects are eosinophilia, infections and dermatitis at the injection site. In both cases, the profile of side effects is not very serious, which represents a comparative advantage with respect to the rest of the systemic treatments commonly used.

There are other biological drugs that could be useful for the treatment of BP, such as RTX an anti-CD20 drug. Although there is data on its effectiveness (with response rates of 70.5%) [22,23], Its side effect profile, which includes the possibility of serious infections and oncological processes, would a priori not make it optimal for the management of patients with BP with multiple comorbidities, which is why its data have not been the objective of this systematic review.

The main limitation of the review presented is the quality of the studies included in it, most of them being case series or individual cases. The development of studies with a higher level of scientific evidence in the near future would be of great interest for adequate knowledge of the degree of effectiveness of omalizumab and dupilumab in BP. Moreover, the lack of standardized criteria for defining clinical response in BP treatment could make it difficult to compare the effectiveness between both treatments. 

## 5. Conclusions

BP patients with treated with omalizumab and dupilumab show a profile of sociodemographic and clinical characteristics that could be comparable to that of patients with BP treated in routine clinical practice, although these are patients refractory to other systemic treatments, including corticosteroids and immunosupppresive agents. Both omalizumab and dupilumab appear to be effective options for the treatment of BP in patients refractory to other pharmacological therapies. Moreover, omalizumab and dupilumab are drugs with a good safety profile for use in patients with BP, with adverse reactions associated with their use being rare and generally mild.

## Figures and Tables

**Figure 1 jcm-13-04844-f001:**
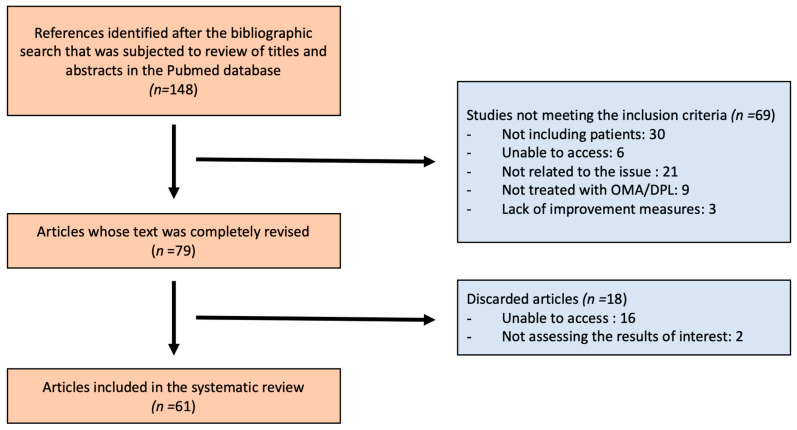
Search strategy. OMA (omalizumab); DPL (dupilumab).

**Table 1 jcm-13-04844-t001:** Overview of data regarding omalizumab and dupilumab treatment for Bullous Pemphigoid.

	Omalizumab	Dupilumab
Number of patients treated	363	523
Approved indications for the drug	Severe allergic asthma, chronic spontaneous urticaria, chronic risnosinusitis with nasal polyps	Asthma, nodular prurigo, atopic dermatitis, chronic rhinosinusitis with nasal polyposis, eosinophilic esophagitis, hand dermatitis
Route of administration and dosage used in the studies	300 mg/450 mg/600 mg subcutaneously every 2–4 weeks	600 mg subcutaneously initially followed by 300 mg subcutaneously every 1–2 weeks
Efficacy data of omalizumab and dupilumab to treat BP	Complete response: 76.13%, improvement in BP between 2 weeks and 6 months after starting treatment	Complete response: 70.39%, healing achieved between 2 weeks and 6 months after starting treatment
Safety data	Less than 1% of patients presented adverse events, the most frequent were intense pruritus that resolved with dupilumab and dermatitis at the injection site that resolved spontaneously.	Less than 1% of patients presented adverse events, the most frequent were dermatitis at the injection site that resolved spontaneously and eosinophilia that resolved by adding IS.

## Data Availability

The original contributions presented in the study are included in the article, further inquiries can be directed to the corresponding author.

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
