# Peer review of "Omalizumab and Dupilumab for the Treatment of Bullous Pemphigoid: A Systematic Review"

_jcm, 2024, doi:10.3390/jcm13164844_

Round 1

Reviewer 1 Report

Comments and Suggestions for Authors

Dear authors

Thank you very much for your manuscript which is an interesting topic and useful for clinical practice. However, I feel that this manuscript should be revised to improve its clarity.

1. What do "AP" or "PA" in this manuscript strand for? I think they are typo. If they referred to bullous pemphigoid or BP, it should be revised accordingly.

2. They are too many acronyms in this article which I think they are difficult to understand. Dupilumab and omalizumab is only one word. They are not necessary to use acronym.

3. Sb in table 1?  Is it subcutaneously?  should be corrected

4. As the effectiveness of dupilumab and omalizumab in BP in this manuscript was derived from case reports and systematic review, how the authors concluded "complete remission/response in omalizumab and dupilumab were 76.13% and 70.39% (Table 1)" 

5. Furthermore, the included articles may define complete remission or complete response differently. From above sentences, they may mislead to readers that omalizumab has higher efficacy than omalizumab in treating BP (Table 1). ???

6. Appendix A should be edited to make it easier to read.

Author Response

Dear Editorial Team and reviewers,

The authors of the manuscript would like to thank you for your comments, as they allow us to improve the scientific quality of our research. All the suggested changes have been implemented. Below, you can see a point-by-point response to your commentaries:

Reviewer 1:

Thank you very much for your manuscript which is an interesting topic and useful for clinical practice. However, I feel that this manuscript should be revised to improve its clarity.

  1. What do "AP" or "PA" in this manuscript strand for? I think they are typo. If they referred to bullous pemphigoid or BP, it should be revised accordingly.

Thank you for your comment. “AP” and “PA” were mistakes. This has been corrected.

  1. They are too many acronyms in this article which I think they are difficult to understand. Dupilumab and omalizumab is only one word. They are not necessary to use acronym.

This acronyms have been modified so as to make the text more easily understandable.

  1. Sb in table 1?  Is it subcutaneously?  should be corrected

This acronyms have been modified so as to make the text more easily understandable.

  1. As the effectiveness of dupilumab and omalizumab in BP in this manuscript was derived from case reports and systematic review, how the authors concluded "complete remission/response in omalizumab and dupilumab were 76.13% and 70.39% (Table 1)" 

This information was obtained calculating the percentage of articles and review who reported a complete improvement of the BP with those treatments. A sentence has been included in the “Methods” section so as to clarify this point.  

  1. Furthermore, the included articles may define complete remission or complete response differently. From above sentences, they may mislead to readers that omalizumab has higher efficacy than omalizumab in treating BP (Table 1). ???

Thank you for your comment. This point is very important, so we have added this as a limitation in the limitations section. It is not possible to compare directly efficacy rates between treatment, as the studies included in the review are not homogeneous.

  1. Appendix A should be edited to make it easier to read.

Thank you for your comment. We have improved the text within the table, making it more homogeneous, so as to make it more easily readable.

Reviewer 2 Report

Comments and Suggestions for Authors

Dear Authors,

I read with interest the manuscript entitled: “Omalizumab and Dupilumab for the treatment of Bullous Pemphigoid: A systematic Review”.

In my opinion this revision is very interesting as is a frequent situation in clinical practice the use of Omalizumab or Dupilumab in elder patients with BP. But the treatment is off-label, and the article provides scientific insights for the use of these therapies in refractory cases.

I procced with my comments:

-Line 108: Point 2.5, and 2.6 and 2.7. Why do you define so many variables that afterwards you don´t include in results? I mean, for example, why do you explain the assessment of the quality of evidence and after there is nothing in relation in results?

-Line 189: Do some authors use maintenance therapy with OMA?

-Table 1: Las line seems a conclusion more than a results, you should specify a number or percentage?

-Discussion: in my vision the discussion is superficial, OMA and DPL can be more deeply evaluated as nowadays are the main alternative in BP, and probably the ones with long-term efficacy an safety.

Author Response

Dear Editorial Team and reviewers,

The authors of the manuscript would like to thank you for your comments, as they allow us to improve the scientific quality of our research. All the suggested changes have been implemented. Below, you can see a point-by-point response to your commentaries:

I read with interest the manuscript entitled: “Omalizumab and Dupilumab for the treatment of Bullous Pemphigoid: A systematic Review”.

In my opinion this revision is very interesting as is a frequent situation in clinical practice the use of Omalizumab or Dupilumab in elder patients with BP. But the treatment is off-label, and the article provides scientific insights for the use of these therapies in refractory cases.

I procced with my comments:

-Line 108: Point 2.5, and 2.6 and 2.7. Why do you define so many variables that afterwards you don´t include in results? I mean, for example, why do you explain the assessment of the quality of evidence and after there is nothing in relation in results?

Thank you for your comments. The assessment of the quality of evidence is collected in the “Appendix Tables”, regarding each article. The rest of the variables are, as well, included in the tables and also are commented in the main text.

-Line 189: Do some authors use maintenance therapy with OMA?

Some authors used long-term dosages of omalizumab, whereas some authors only reported “classic” dosage of OMA for chronic spontaneous urticaria. This information has been included in the results section.

-Table 1: Las line seems a conclusion more than a results, you should specify a number or percentage?

Thank you for your comment. Last line of table 1 has been deleted to make the table more appropriate for the “results” section.

-Discussion: in my vision the discussion is superficial, OMA and DPL can be more deeply evaluated as nowadays are the main alternative in BP, and probably the ones with long-term efficacy an safety.

Thank you for your comments. We have extended the discussion with novel data so as to make it more complete.

Round 2

Reviewer 1 Report

Comments and Suggestions for Authors

Dear authors

There are still some typo. 
Abstract : AP should be BP. Line 16-17 : omalizumab and Dupilumab are repetitive which are similar to line 61,66.

Author Response

Thank you for your comments. The mistakes have been corrected